# *body2vec:* 3D Point Cloud Reconstruction for Precise Anthropometry with Handheld Devices

**DOI:** 10.3390/jimaging6090094

**Published:** 2020-09-11

**Authors:** Magda Alexandra Trujillo-Jiménez, Pablo Navarro, Bruno Pazos, Leonardo Morales, Virginia Ramallo, Carolina Paschetta, Soledad De Azevedo, Anahí Ruderman, Orlando Pérez, Claudio Delrieux, Rolando Gonzalez-José

**Affiliations:** 1Laboratorio de Ciencias de las Imágenes, Departamento de Ingeniería Eléctrica y Computadoras, Universidad Nacional del Sur, and CONICET, Bahía Blanca B8000, Argentina; pnavarro@cenpat-conicet.gob.ar (P.N.); bpazos@cenpat-conicet.gob.ar (B.P.); lmorales@cenpat-conicet.gob.ar (L.M.); cad@uns.edu.ar (C.D.); 2Instituto Patagónico de Ciencias Sociales y Humanas, Centro Nacional Patagónico, CONICET, Puerto Madryn U9120, Argentina; ramallo@cenpat-conicet.gob.ar (V.R.); paschetta@cenpat-conicet.gob.ar (C.P.); deazevedo@cenpat-conicet.gob.ar (S.D.A.); ruderman@cenpat-conicet.gob.ar (A.R.); orlandoperez@cenpat-conicet.gob.ar (O.P.); rolando@cenpat-conicet.gob.ar (R.G.-J.); 3Departamento de Informática, Facultad de Ingeniería, Universidad Nacional de la Patagonia San Juan Bosco, Trelew U9100, Argentina

**Keywords:** deep learning, neural networks, structure from motion, 3D point cloud, anthropometry

## Abstract

Current point cloud extraction methods based on photogrammetry generate large amounts of spurious detections that hamper useful 3D mesh reconstructions or, even worse, the possibility of adequate measurements. Moreover, noise removal methods for point clouds are complex, slow and incapable to cope with semantic noise. In this work, we present body2vec, a model-based body segmentation tool that uses a specifically trained Neural Network architecture. Body2vec is capable to perform human body point cloud reconstruction from videos taken on hand-held devices (smartphones or tablets), achieving high quality anthropometric measurements. The main contribution of the proposed workflow is to perform a background removal step, thus avoiding the spurious points generation that is usual in photogrammetric reconstruction. A group of 60 persons were taped with a smartphone, and the corresponding point clouds were obtained automatically with standard photogrammetric methods. We used as a 3D silver standard the clean meshes obtained at the same time with LiDAR sensors post-processed and noise-filtered by expert anthropological biologists. Finally, we used as gold standard anthropometric measurements of the waist and hip of the same people, taken by expert anthropometrists. Applying our method to the raw videos significantly enhanced the quality of the results of the point cloud as compared with the LiDAR-based mesh, and of the anthropometric measurements as compared with the actual hip and waist perimeter measured by the anthropometrists. In both contexts, the resulting quality of body2vec is equivalent to the LiDAR reconstruction.

## 1. Introduction

Reconstruction of 3D objects is among the many potential applications of Computer Vision models working together with Deep Learning techniques. This combined approach can contribute to solve common problems regarding the analysis of human body shape. In several contexts, including sports, health, and other contexts, there is a recurrent need to have accurate anthropometric measurements (i.e., the shape, form, size, and several perimeters and volumes of the human body) [1,2,3]. This is the case in many clinical applications ranging from diagnostic, treatment, and follow-up of overweight-related conditions, to less frequent but important skeletal pathologies, such as scoliosis [4]. Obesity-related conditions constitute a specifically critical case, since overweight and obesity have become increasingly widespread, considered one of the main public health challenges of the 21st century [5]. Overweight is diagnosed and clinically treated after the assessment of anthropometric traits including weight, height, and several body perimeters [6]. These measurements are typically obtained by traditional manual methods, which are imprecise, require specific professional intervention, and may turn to be incomplete. For instance, in diagnosing obesity, a key indicator is the distribution of abdominal adipose tissue, which is an aspect of geometric shape rather than a relationship among classical anthropometric measures [7].

The acquisition of complete 3D models of human bodies, and its translation into data adequately represented for clinical and non-clinical practices, encloses several difficulties. First, even when the person can be constrained to remain motionless inside a full body scanner device, incomplete surface data is obtained generated by occlusions [8]. This leads to a loss of quality due to missing data in the occluded areas. Second, given their cost and complexity of use, short-range LiDAR scanners are still out of the scope of most physicians, specialists, and research groups. Finally, even though traditional anthropometric measurements are acknowledged to be imprecise in the assessment of overweight-related conditions, they still represent a cheap and easy way to gather information regarding the weight condition of an individual [9,10,11,12,13].

The study of human body shape needs to evolve from the classical somatotype/anthropometric approaches towards inexpensive and practical 3D technologies, and data in digital format. Apart from overweight related issues, clinical applications of 3D body scanning include the precise diagnosis of skeletal pathologies, such as scoliosis, and prosthetic design, among others. Non-clinical applications of 3D body scanning are also diverse. Bodybuilding, fitness, and high competition performance would largely benefit from an accurate and straightforward method to register and visualize body shape and its evolution throughout specific training or activities. Online outfit sales also can be enhanced by replacing the classical size conventions by using individual-specific avatars as the basis to select the proper outfit model and size. In addition, the ability to register data in digital format will enable novel scientific pursuits, for instance to develop population-wise body shape studies (e.g., stratified by ethnic group, age, geographical location, nourishing habits, etc.). Collectively, this would trigger a more refined capture of geometric data, better measurements, and full potential to compile large datasets in the screening of diverse populations. This will in turn enable a rapid, precise, and non-invasive quantification of human body shape [14,15,16].

Given the diversity of applications and the associated intrinsic complexity, body shape analyses require innovative technological approaches in order to improve the accuracy and precision in the acquisition, processing and analysis of digitized data. Among the first human body acquisition and tracking approaches, we can mention typical feature-engineering-based methods. Mikik et al. [17] for instance, applied body part localization procedures based on template fitting and region growing. These templates are computed from 2D silhouettes, using prior knowledge of average body part shapes (ellipsoids and cylinders) and dimensions using synchronized multiple cameras. Later, more elaborate generative methods enabled the automatic recovery of human shape and pose from images, which leverages learned deformation models using template meshes (from scanners) for graphics applications [18].

More recently, 3D surface scanners provided automated and accurate measurements of body shape adequate for clinically and anthropometric applications. Ng et al. [19] applied more sophisticated techniques based on anatomical landmark positioning, and measurements of idealized circumferences, areas, and volumes, which allow an assessment of the bodily fat distribution and its relationship with metabolic disorders, body mass index, other anthropometric indexes, and their relation to within and among ethnic groups diversity. In addition, in some applications in health and medicine, it is necessary to segment body parts or volumes (arms, legs, head, torso) from 3D scanned data [20]. For this, fitting techniques are used to deform template models to recognize segment endpoints, determine their locations, and take measurements of them. In others applications, like support tools for fitness, exercise guidance, and wellness activities, it is common to use off-the-shelf 3D devices for body scanning, for example, Microsoft Kinect^®^. This allows a user to obtain morphometric data to evaluate the body shape and physical conditions of trainees with acceptable precision [21].

Close-range photogrammetry based on structure from motion (SfM) recently emerged as a viable alternative in several applications, ranging from industrial context to cultural heritage preservation, engineering, Earth sciences, and several other 3D imagery tasks. The photogrammetric procedure typically takes successive frames in a video, determines the correspondence between salient points in each frame, and infers the extrinsic camera parameters with which the actual 3D position of these points can be determined. SfM is advantageous for several reasons: it is low-cost, flexible, only requires widespread acquisition devices (for example smartphones or tablets) [22]. This allows quick and easy scanning and point cloud generation. These point clouds, when are computed from adequate video takes, may have only few spurious points, allowing useful characterization and geometric measurements in geographic and urban scales [23]. However, in very close range acquisition, as is the case in 3D body scanning and reconstruction, plain SfM is unable to achieve the precise and high-quality geometry required to obtain accurate measurements. This is mostly due to noise in the background that generate spurious photogrammetric determinations, which in turn deliver wrongly detected points that alter significantly the quality of the point cloud. In this condition, 3D reconstruction and geometric measurements with a noisy point cloud will produce results that are upright unusable. Further decimation and filtering algorithms are inadequate in this context since they distort the underlying body structure, thus leading to inaccurate 3D models. This situation can be mitigated by carefully using a clear background, even illumination, and high-end devices, all conditions that go against the very simplicity and inexpensiveness of SfM in other contexts.

In this work, we present body2vec, a model-based approach to background filtering in 3D body scanning. Our method pre-processes the acquisition frames using a convolutional neural network (CNN) that identifies the region of interest (human silhouettes) and filters out the background, thus leading to much cleaner point clouds and more precise subsequent meshes. The final result is both robust with respect to acquisition conditions (background, illumination, and video quality) and accurate enough to produce quality anthropometric measurements. Thus it can be successfully applied to videos taken with low cost devices like smartphones or tablets. Applying our method resulted in an error reduction of almost an order of magnitude (measured as the mean absolute distance to the LiDAR mesh taken as a silver standard). Finally, we estimated the hip and waist perimeters using a very simple fit, comparing the results of the LiDAR-based mesh with body2vec against the actual anthropometrists’ measurements taken as gold standard, achieving 1.23 cm less mean error in average in the hip measurement and 3.21 cm higher error in the waist measurement.

## 2. Materials and Methods

In this section, (i) we describe the data collection in detail, including raw video takes from smartphones, LiDAR based, and anthropometric; (ii) we present BRemNet (Background Removal Network), a human body identification and segmentation model that is applied for video background removal; (iii) we generate SfM point clouds from the raw and clean videos, generate a registration thereof to the meshed LiDAR acquisition, and measure the respecive registration errors; and (iv) we evaluate anthropometric measurements from the LiDAR mesh, and the raw and clean point clouds, and compare them to the measurements performed by anthropometrists. An overview of the complete process can be seen in Figure 1.

### 2.1. Data Collection

Smartphone videos and 3D body scans were taken from 60 volunteers (38 females, 22 males; average age = 39; sd = 12) within the facilities of the Puerto Madryn Regional Hospital. All subjects gave their informed consent for inclusion before they participated in the study. The study was conducted in accordance with the Declaration of Helsinki, and the procedure was approved by the Ethics Committee of the Puerto Madryn Regional Hospital under protocol number 19/17 (approved 4 September 2018). Even though higher quality SfM reconstructions could be obtained with high resolution photography, in this context this would be inadequate given the acquisition time required. However, given the rapid evolution of smartphones, it is foreseeable that very high resolution video will be feasible in the near future. It is worth noting that the problem addressed in this paper is not concerned with low resolution acquisition, but of low quality SfM reconstruction due to semantic noise, which is not related to resolution.

Videos were recorded in a single take, completely surrounding the volunteer while they stood in underwear or tight clothes with their arms extended and legs shoulder-width apart (see a typical frame in Figure 2a). The takes were about 35 s long, in MPEG-4, 1920×1080 @ 30 fps. At the same time, a 3D body scan was obtained using the first version of the Structure™sensor scanner (more detailed description of this data collection are published in Reference [24]). This latter acquisition generates a high-quality point cloud and subsequent 3D mesh, which will be taken as the reference for our video-based 3D reconstruction. Structure™sensor scanner was our best choice to achieve LiDAR quality with a handheld device able to perform quick captures, with harmless sensing technology, and an affordable price. Finally, anthropometric measurements were acquired by trained domain experts using the standard protocol, including total height (using the Seca 206 mechanical measuring tape, Seca GmBH & Co Kg, Hamburg, Germany), total weight, and body composition (muscle mass, fat-free, and body fat mass and percentages) estimated with a bioimpedance scale (Tanita BC 1100F), and hip and waist circumferences, using ergonomic measuring tape Seca 201 (Seca GmBH & Co Kg, Hamburg, Germany).

### 2.2. Segmentation Model

As already stated, close-range photogrammetry based on SfM will likely generate low quality point clouds which will require extensive subsequent filtering that will hamper the geometric accuracy of the acquisition. Our strategy is to perform a background removal in all the video frames prior to applying SfM, taking into advantage the fact that the foreground is always a human figure, which implies that a specific semantic segmentation model can be developed using machine learning. We used the *Mask R-CNN* architecture as a baseline for identification and segmentation of human bodies. This method attempts to identify the pixel-level regions of each body instance in an image. In contrast to semantic segmentation, instance segmentation not only distinguishes semantics, but also different body instances. The model was trained to learn a pixel-level mask for a single class person. Below, we describe the architecture and functioning of the underlying model.

Mask R-CNN is a fully convolutional network (FCN) designed to help locate objects at pixel level and for semantic segmentation [25]. The underlying model is optimized over prior proposals for a multi-task loss function that combines the losses of classification, bounding box localization and segmentation mask L = $L_{\mathrm{cls}}$ + $L_{\mathrm{box}}$ + $L_{\mathrm{mask}}$. $L_{\mathrm{cls}}$ and $L_{\mathrm{box}}$. The loss function encourages the network to map each pixel to a point in feature space in a way such that pixels belonging to the same instance lie close together, while different instances are separated by a wide margin Reference [26]. $L_{\mathrm{mask}}$ is defined as the average binary cross-entropy loss, only including the *k*-th mask if the region is associated with the ground truth class *k*, where $Y_{ij}$ is the label of a cell (i,j) in the true mask for the region; ${\hat{y}}_{ij}^{k}$ is the predicted value of the same cell in the mask learned for the ground truth class *k* (see Equation (Equation 1)).
(1)$$L_{\mathrm{mask}}=-\frac{1}{m^{2}}\sum_{1\leq i,j\leq m}\left[y_{ij}\log{\hat{y}}_{ij}^{k}+\left(1-y_{ij}\right)\log\left(1-{\hat{y}}_{ij}^{k}\right)\right].$$

We developed BRemNet, a further refinement of Mask R-CNN with the aim of pre-processing per frame the videos taken specifically for photogrammetric 3D body reconstruction (see Figure 3). As with Mask R-CNN, we use an RPN, but we add a binary classifier, and a background removal and chroma coding step. The RPN generates a set of bounding boxes which may contain the human body within the video frame. These boxes are refined using the Mask R-CNN regression model (see Figure 2a). The binary classifier was trained to label pixels as foreground/background using the pre-trained weights of the Microsoft Common Objects in COntext (MS COCO) [27] containing the labeled person. We prepared a training dataset with 200 frames with different bodies in different frame locations. These frames were manually annotated using the VGG Image Annotator [28]. The result of this step is then a binary mask containing the silhouette of the human body present in the frame (see Figure 2b). The mask is used in the final background removal and chroma coding step (see Figure 2c). After this processing, the video takes are converted to a set of about 500 frames in which the foreground (the human body) remained unchanged and the background was set into green, which reduces the error introduced in the subsequent SfM step.

### 2.3. 3D Reconstruction and Measurement

Structure from Motion photogrammetry provides point clouds computed from multi-views or video takes. These point clouds in some cases can be close in quality to those generated by LiDAR sensors. For this reason, their use is steadily gaining popularity both in long range and in close range applications. Recently, several open-source libraries and applications were released to process SfM in different contexts. In particular, Visual SFM [29] implements the specific case of frame sequence matching, which is adequate for a frame sequence from a video. This procedure is performed both with the original video takes, and with the same takes pre-processed by the segmentation model described in the preceding subsection.

Our long term goal is to extract highly accurate and precise anthropometric measurements from the point clouds generated from videos taken with smartphones or similar devices. In this work, we focus on the abdominal perimeter, which is one of the most representative values related to overweight and similar conditions that currently require frequent and precise assessment in large populations given the current obesity epidemics. For this, we expect to determine the accuracy and precision of the SfM-based assessment, as compared with the LiDAR-based and with the direct measurements performed by trained anthropometrists. The procedure to this avail is to select the points that correspond to the navel height in the subject (the place where the anthropometrists take the actual abdominal perimeter), fit these points to an ellipse, in which the perimeter is the final estimation produced by the model. This navel-height point selection, ellipse fitting, and perimeter measurement is performed on the three point clouds available for the same individual (LiDAR-based, unprocessed video take, processed video take).

## 3. Results

Below, we evaluate the 3D body reconstruction quality results achieved using BRemNet for masking the human silhouette in the video takes. First, we assess the quality of the segmentation mask with respect to manually segmented masks. Then, we evaluate the enhancements with respect to the point clouds generated using SfM from the raw videos. Finally, we compare the accuracy and precision of the point clouds obtained after masking with respect to LiDAR-based when proposed model is employed, using the anthropometrist’s measurements as a gold standard.

### 3.1. Mask Segmentation

Four quality metrics were applied to evaluate pixel-wise segmentation using BRemNet: Hamming Loss metric, Jaccard index, F1-measure, and Accuracy, against manually segmented masks on 20 frames randomly chosen. We considered the RoI determined by the minimax rectangle of the manually segmented mask in each case. Pixel positive condition then arises when pixels belong to the manual mask, and thus true and false positives, and true and false negatives are defined accordingly. In addition, we compared these results of BRemNet model with the segmentations generated by Mask R-CNN (see Table 1).

Even though BRemNet performed better than Mask R-CNN in all the quality metrics, the improvement is only marginal. However, the true advantage of BRemNet is related to the consistency of the resulting mask during all the frames of a given take. The segmentation quality of Mask R-CNN is strongly dependent on the cleanness of the background and the stability of the take. For instance, some objects in the background together with part of the actual subject can be misleadingly identified as another object (e.g., a dog), resulting in a frame with a higher amount of false negatives.

Similar situations arise when the subject moves during a take, or if the camera movement along the subject is uneven. We explored the per-frame differences among BRemNet and Mask R-CNN in ten randomly chosen videos, using Jaccard index. The proportion of frames with a similarity less than 0.8 ranges from 3.23% to 38.57%, and the Jaccard index of the least similar frames can drop down to 0.37 (see Table 2 and Figure 4). Although not significant in the average, these Mask R-CNN bad frames result in a poor SfM point cloud reconstruction, since parts of the photogrammetric information will be wrongly inferred. In these takes, we selected the frames with larger disparity between Mask R-CNN and BRemNet (in this case the minimum Jaccard index among both segmentations). In these ten frames (one for each video), we manually segmented the expected mask and established the quality measures of Mask R-CNN and BRemNet (see Table 3). This analysis confirms that Mask R-CNN is prone to worst cases that may hamper the resulting point cloud, while BRemNet performs in a much more stable manner.

### 3.2. Segmented Point Cloud Evaluation

As mentioned in Section 2.3, the frames segmented by BRemNet were used to compute clean point clouds using SfM. On average, these point clouds are composed of 37.265 points each. This represents almost an order of magnitude less than the size of the resulting point clouds computed with the raw video (i.e., without prior use of BRemNet), which in average were 229.659 points in size. On the other hand, LiDAR-based mesh models were previously processed with Laplacian smoothing and hole closing using the algorithms presented in Reference [30]. We thus obtained three full-body 3D models, the LiDAR-based mesh, and the raw and clean SfM-based point clouds (see Figure 5). Assuming the LiDAR-based as a silver standard, we used CloudCompare [31] to compare the raw and clean point clouds to the mesh. We measured Root Mean Square Error (RMSE), Mean Distance, and Standard Deviation using Iterative Closest Point (ICP) as registration method, after the proper alignment between the mesh and the point cloud using PCA (see Figure 6). As a registration method, ICP implements nearest neighbour reconstruction using Euclidean distance to estimate the closest point between two entities. Since it is an iterative process, the registration error slowly decreases during this procedure.

We computed the Root Mean Square Error (RMSE) between the mesh and the two point clouds in the 60 3D models. The mean RMSE of the raw point cloud was 12.11 cm, while error was reduced in the clean cloud to 2.02 cm. We also computed the Mean Distance (MD), i.e., the mean of distances between each point in the cloud to the nearest triangle in the mesh. In the raw point cloud the MD was 6.28 cm, while, in the clean cloud, the MD dropped to 0.04 cm. Finally, we computed the Standard Deviation (SD) of the MD, which in the raw point cloud was 10.4 cm and in the clean one was 1.9 cm. All these results are shown for the 60 full-body reconstructions in Figure 7.

### 3.3. Abdominal Perimeter Measurements

With the three resulting full-body reconstructions, we performed two anthtropometric measurements, namely hip and waist perimeters. Along with body-mass index, hip and waist perimeters constitute the most widely anthropometric measurements used to detect and analyze overweight conditions. These results were then compared to the actual measurements taken by anthropometrists with the traditional instruments. First, the 3D model is scaled to the measured anthropometric height of the dataset and the centroid is obtained. Around the mean height (on the Y axis), an RoI is considered that takes slices of 1 cm height and orthogonal to the Y axis (i.e., parallel to the floor). Points within each of the slices are fitted to ellipses [32], of which we calculate the perimeter using the Ramanujan approximation. According to the anthropometrists’ practice, the hip is the region with the largest perimeter slightly below the middle height of the subject. With our procedure, in the three full-body reconstructions we searched the slice in which this condition arise, and use the perimeter of the fitted ellipse as a predictor of the hip perimeter. In the waist, however, there is no single consensus as to where the actual perimeter should be measured in the subjects, which is usually taken slightly below the navel. Lacking this specific shape trait, we adopted a criterion which is to search the slice above the middle height of the subject whose fit was the closest to the actual anthropometrist’s measurement (see Figure 8).

We evaluated the error of these two measurements for the three full-body reconstructions against the actual measurements performed by anthropometrists. The absolute error was calculated from each waist and hip measurement respect to measurement taken by anthropometrists. In Table 4, we show the mean and SD of the two measurements in the 60 subjects. Finally, we performed a least squares regression between the actual anthropometrists’ measurements and the estimated measurements performed on the clean point cloud computed with the BremNet-filtered videos (see Figure 9).

## 4. Discussion and Conclusions

We performed geometric reconstructions of the waist and hip of 60 subjects, for which we have also the actual anthropometric ground truth, for both the LiDAR-based and the masked SfM-based point cloud. Our final goal was to assess the applicability of 3D body reconstruction based on handheld video acquisition for anthropometric purposes. The proposed approach increases significantly the quality and robustness of the SfM-point cloud. Masked point clouds have 83.31% less RMSE, 99.32% less mean distance, and 81.72% less standard deviation as compared with the LiDAR-based silver standard. Regressions of both anthropometric measures using both technologies over the 60 subjects show an equivalently acceptable quality. Hip reconstruction is less accurate, since both LiDAR and SfM acquire a tight anatomic surface of the subjects, while the anthropometrists’ tape evens out the buttocks separation. This aspect is currently being considered in our reconstruction model.

These preliminary results are quite promising, since there is still room for optimization. In particular, the BRemNet model used transfer learning from a general purpose machine vision net, but it can be retrained with a larger set of manually segmented masks to have a higher accuracy than the one shown in Figure 4. In addition, as mentioned above, for hip and waist perimeters (and other anthropometric measures) a better anatomical model can lead to better estimations than the one used in this paper. Finally, with a larger sample set, a more stable and refined regression model can be established to yield final estimations that takes into account other aspects of the subjects’ information apart from the acquisition (e.g., somatotype, gender, ethnicity, etc.).

This contribution is related to achieve useful *measurements* only, not to produce 3D reconstruction of good quality, which are clearly better performed with LiDAR acquisition. However, these measurements can be used to feed quite realistic and personalized avatars (e.g., using the computational bodybuilding model), triggering a significant set of possible applications in human health, online outfit retail, and sports, to mention just a few. In particular, as already presented elsewhere [33], it will be possible to verify that the morphometry of the human body is a robust predictor of biomedical phenotypes related to conditions, such as obesity and overweight, as well as validating its usefulness in routine clinical practice.

## Figures and Tables

**Figure 1 jimaging-06-00094-f001:**
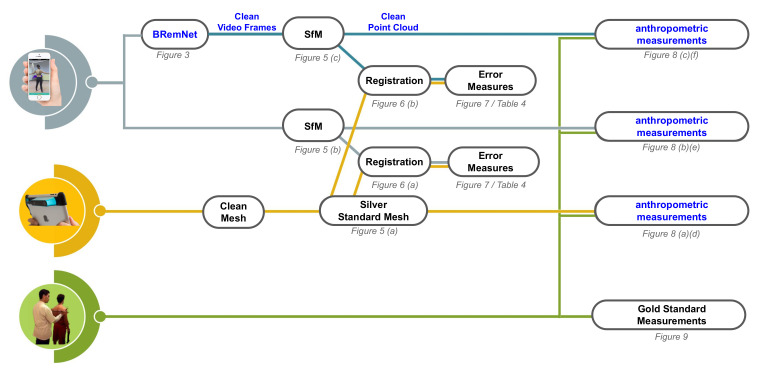
Overview of workflow. Inputs: Raw video in gray, LiDAR-Scanner in yellow, and Classical anthropometry in green.

**Figure 2 jimaging-06-00094-f002:**
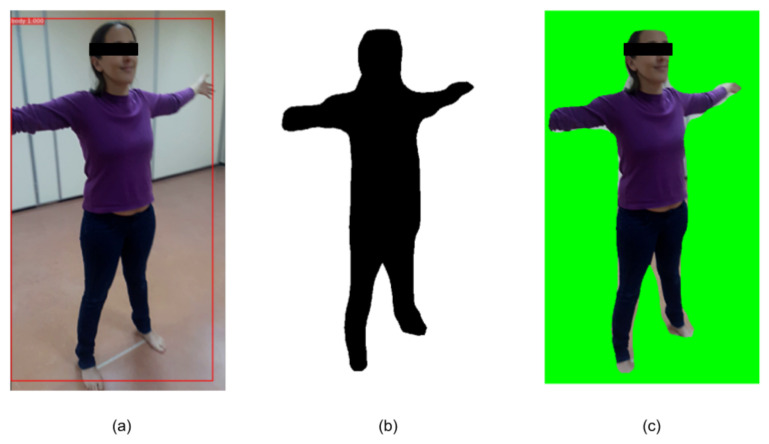
Intermediate results of BRemNet. (**a**) Bounding box, (**b**) Mask, and (**c**) Chroma.

**Figure 3 jimaging-06-00094-f003:**
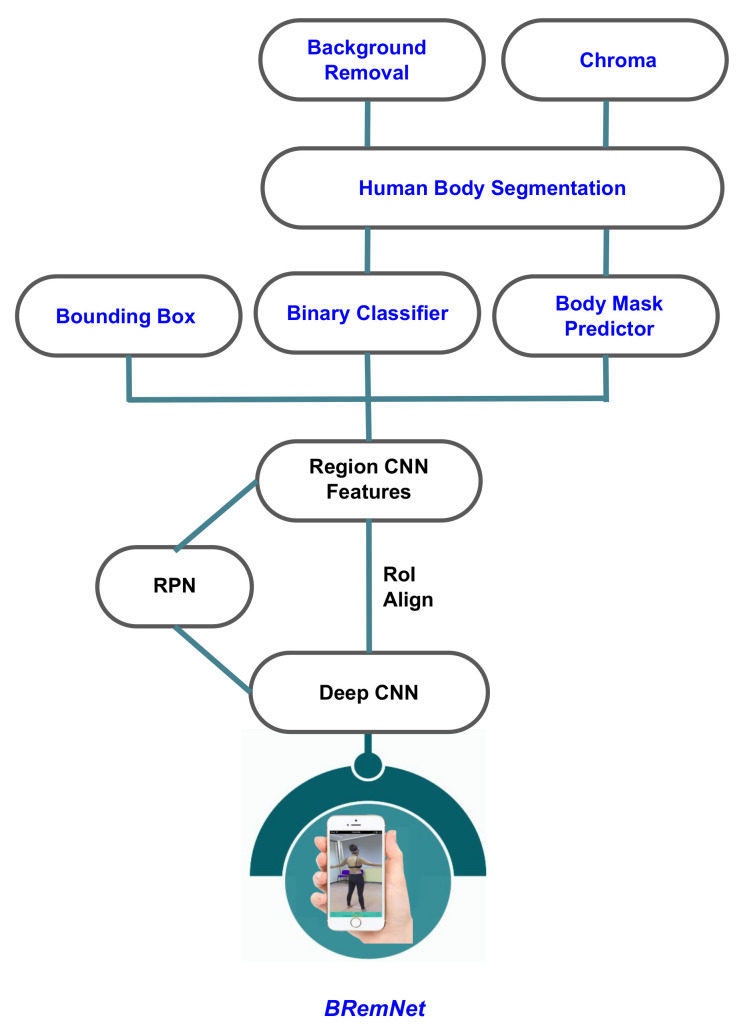
BRemNet architecture.

**Figure 4 jimaging-06-00094-f004:**
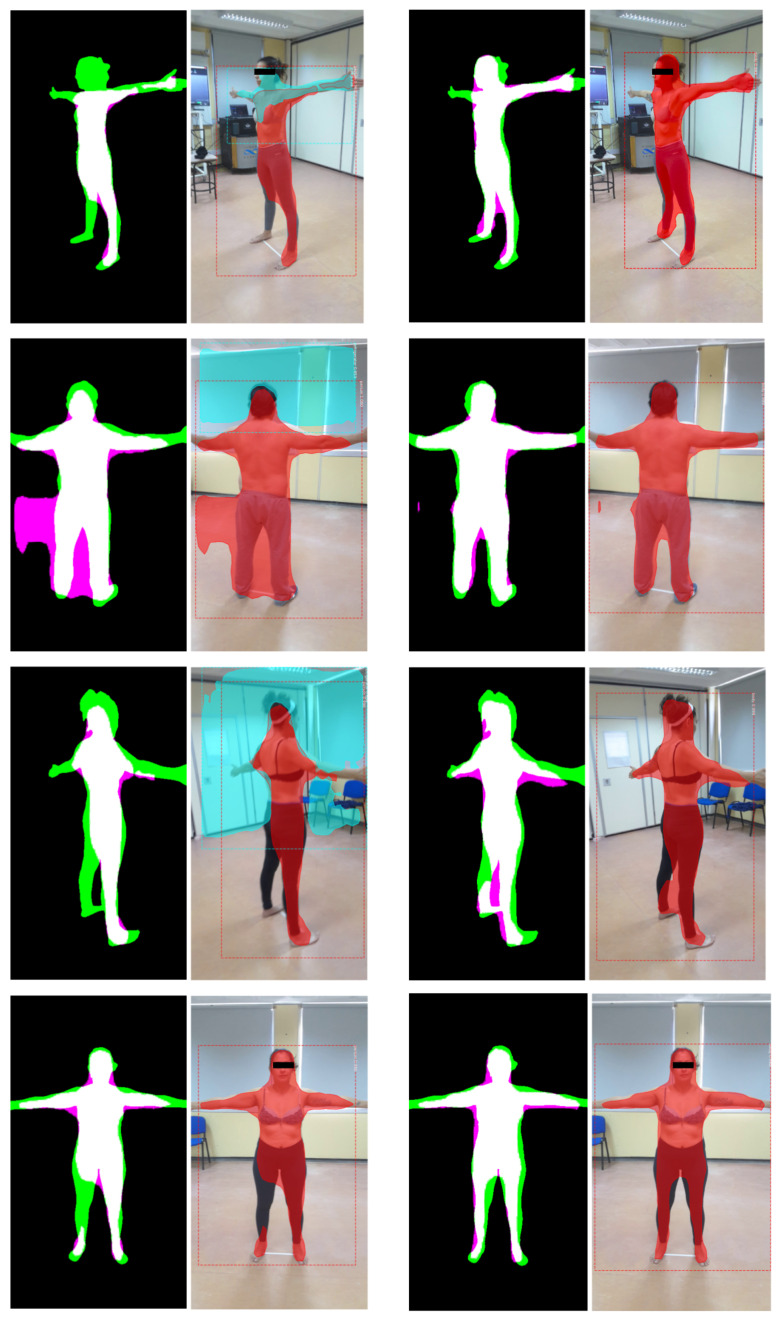
Segmentation examples: Mask R-CNN (left) and BRemNet (right). The segmented mask is superimposed in red to the actual frames. In Mask R-CNN, masks of other identified objects are superimposed in cyan. In masks, true positives are in white, false positives are in magenta, and false negatives are in green.

**Figure 5 jimaging-06-00094-f005:**
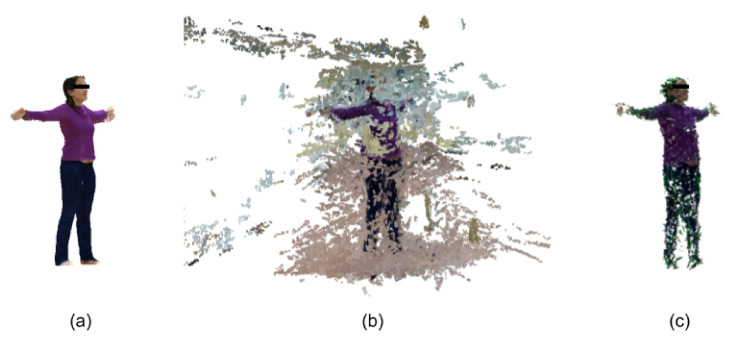
The three 3D full-body reconstructions: (**a**) LiDAR-based mesh cropped, (**b**) structure from motion (SfM)-based point cloud from raw video, and (**c**) point cloud from clean BRemNet-filtered video.

**Figure 6 jimaging-06-00094-f006:**
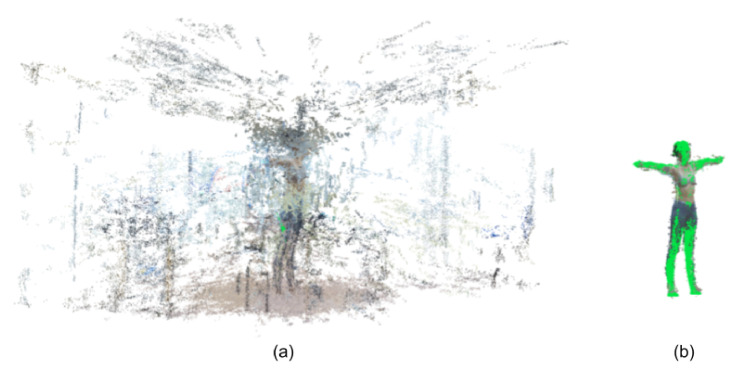
Automatic registration between (**a**) SfM-based point cloud from raw video and (**b**) BRemNet-segmented point cloud with LiDAR-based mesh (green).

**Figure 7 jimaging-06-00094-f007:**
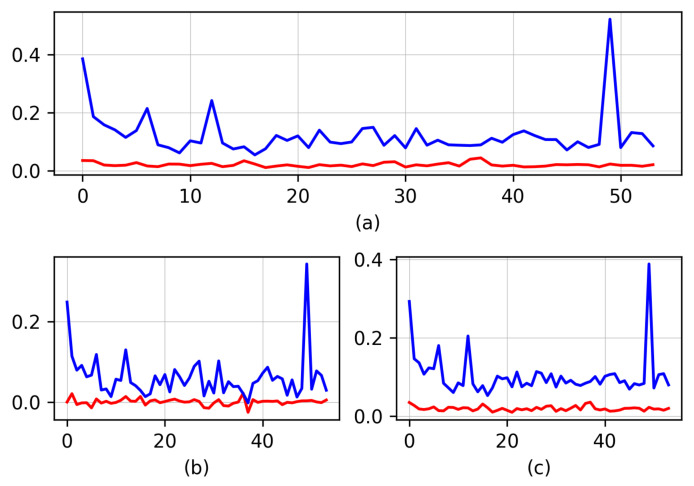
Comparison between the raw point cloud (blue) and the clean one (red) against the LiDAR-based mesh. (**a**) Root Mean Square Error (RMSE), (**b**) Mean Distance, and (**c**) Standard Deviation, all measurements in cm. The x axis represents the volunteer number.

**Figure 8 jimaging-06-00094-f008:**
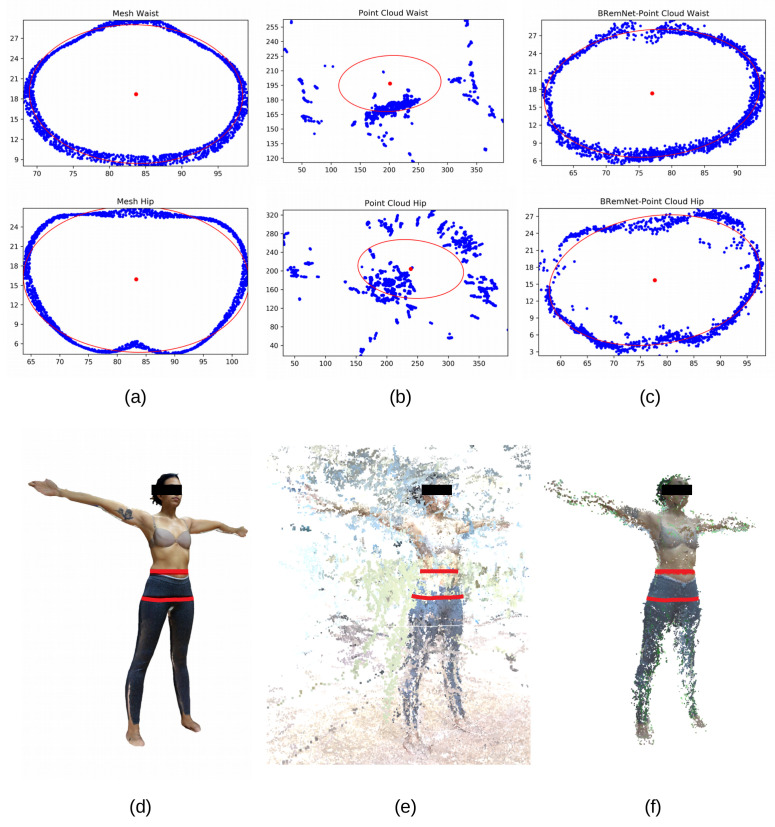
Waist and hip approximation. (**a**,**d**) LiDAR-based mesh, (**b**,**e**) Unsegmented point cloud (scale 1:6), (**c**,**f**) BRemNet-segmented point cloud.

**Figure 9 jimaging-06-00094-f009:**
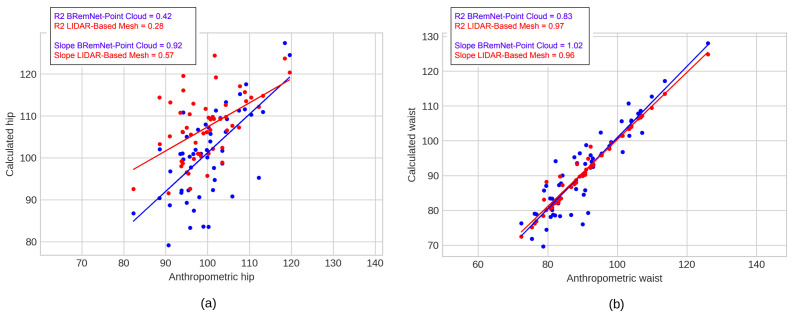
Linear regression of the estimated waist and hip against the actual measurements. BRemNet-Point cloud in blue and LiDAR-based mesh in red. (**a**) Waist and (**b**) hip.

**Table 1 jimaging-06-00094-t001:** Mask segmentation metrics.

Measure	Mean		Standard Deviation		Min		Max	
	BRemNet	Mask R-CNN	BRemNet	Mask R-CNN	BRemNet	Mask R-CNN	BRemNet	Mask R-CNN
Hamming loss	0.04149	0.04734	0.00559	0.00693	0.03428	0.03889	0.05578	0.06111
Jaccard	0.86457	0.84577	0.01913	0.02830	0.83373	0.80468	0.90798	0.89994
F-measure	0.92726	0.91620	0.01096	0.01655	0.90933	0.89177	0.95177	0.94733
Accuracy	0.95851	0.95266	0.00559	0.00693	0.94422	0.93889	0.96572	0.96111
FPR	0.02929	0.03226	0.04308	0.04979	0.03844	0.03938	0.03918	0.04252
FNR	0.07050	0.08385	0.09068	0.11291	0.05415	0.06017	0.06929	0.08992

**Table 2 jimaging-06-00094-t002:** Mask R-CNN vs. BRemNet video segmentation test (*n* = 10).

	Video 1	Video 2	Video 3	Video 4	Video 5	Video 6	Video 7	Video 8	Video 9	Video 10
min Jaccard	0.6000	0.6606	0.7035	0.5057	0.5942	0.5967	0.3723	0.6195	0.4757	0.4541
Jaccard < 0.8	25.57%	11.11%	3.23%	29.73%	12.24%	30.43%	21.43%	21.16%	34.56%	38.57%
max FN	110,746	67,498	77,635	96,063	254,296	59,932	152,988	59,032	87,076	84,309

**Table 3 jimaging-06-00094-t003:** Mask segmentation metrics in the frames with min Jaccard.

Measure	Mean		Standard Deviation		Min		Max	
	BRemNet	Mask R-CNN	BRemNet	Mask R-CNN	BRemNet	Mask R-CNN	BRemNet	Mask R-CNN
Jaccard	0.73543	0.28999	0.11578	0.13299	0.51644	0.06965	0.84800	0.45189
F-measure	0.84264	0.43349	0.08188	0.17417	0.68112	0.13023	0.91775	0.62249
FPR	0.15062	0.50837	0.17849	0.50000	0.10318	0.56022	0.13559	0.51065
FNR	0.15077	0.63428	0.17222	0.74144	0.09727	0.74409	0.13376	0.73075

**Table 4 jimaging-06-00094-t004:** Waist and hip measurement error.

	Mean Error (cm)		Standard Deviation (cm)	
	Hip	Waist	Hip	Waist
LiDAR-based meshes	7.935	**0.910**	6.864	**1.808**
Unsegmented point clouds.	271.708	302.718	87.375	123.548
BRemNet-segmented point clouds	**6.701**	4.128	**4.419**	3.148

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
