# Peer review of "body2vec:* 3D Point Cloud Reconstruction for Precise Anthropometry with Handheld Devices"

_2313-433X, 2020, doi:10.3390/jimaging6090094_

Round 1
Reviewer 1 Report
the paper is well structured and well and the methods are adequately described. The pipe line follow the scientific one and is supported by the result.
I have just one question: why did you compare the body2vecsystem with the raw videos and not with raw photos? I think that you well know that the resolution is really different and that the standard deviation parameters are complete changed.
well, the paper is acceptable but there is just only this remark. It would have been much more innovative if they had conducted the comparison with photos rather than videos
Author Response
Reviewer's comment:
I have just one question: why did you compare the body2vecsystem with the raw videos and not with raw photos? I think that you well know that the resolution is really different and that the standard deviation parameters are complete changed.
Reply:
We are grateful with the reviewer for raising this issue that needs to be explained. We added the following sentences in the first paragraph of Section 2.1 Data Collection, which we think makes clear the issue:
Even though higher quality SfM reconstructions could be obtained with high resolution photography, in this context this would be inadequate given the acquisition time required. However, given the rapid evolution of smartphones, it is foreseeable that very high resolution video will be feasible in the near future. It is worth noting that the problem addressed in this paper is not concerned with low resolution acquisition, but of low quality SfM reconstruction due to semantic noise, which is not related to resolution.
Reviewer 2 Report
This work proposes a new deep learning model, called BRemNet, which is a further refinement of Mask R-CNN. It aims of pre-processing frames of the videos recording the subjects to measure human body. They compare their proposed method to the conventional photogrammetry and LIDAR measured ground-truth data. As the results show, their pre-processing photogrammetry successfully demonstrates a better measurement capability of the human body (waist and hip) compared to that of the photogrammetry without clean the video frames using BRemNet.
The following comments are for the authors to improve their work:
- The authors use the LIDAR-measured mesh as the ground-truth data to do the registration (ICP) with SfM-based and the BRemNet-segmented-based meshes. Could the authors briefly introduce the Iterative Closest Point (ICP) method (2 or 3 sentences are enough)? Also, if they use software to do the registration, could they cite which software they used?
- There are many different 3D scanners, especially for low-cost ones. Why do the authors use Occipital Structure Sensor to do the scanning as the ground-truth data? Could they briefly explain why use this scanner in Section 2.1? Also, there is a new version of Occipital Structure Sensor, called MARK II. Do use this one or the old one?
- What does the BRemNet stand for?
Author Response
Reviewer's comment 1:
Could the authors briefly introduce the Iterative Closest Point (ICP) method (2 or 3 sentences are enough)? Also, if they use software to do the registration, could they cite which software they used?
Reply:
Thanks for pointing this omission. We added the following sentences at the end of the penultimate paragraph of Section 3.2 where ICP is cited:
As a registration method, ICP implements nearest neighbour reconstruction using Euclidean distance to estimate the closest point between two entities. Since it is an iterative process, the registration error slowly decreases during this procedure.
In the same paragraph we mention that the registration software we used is Cloud Compare (with the corresponding citation).
Reviewer's comment 2:
There are many different 3D scanners, especially for low-cost ones. Why do the authors use Occipital Structure Sensor to do the scanning as the ground-truth data? Could they briefly explain why use this scanner in Section 2.1? Also, there is a new version of Occipital Structure Sensor, called MARK II. Do use this one or the old one?
Reply:
We added in the second paragraph of section 2.1 Data Collection the following sentence:
Structure sensor scanner was our best choice to achieve LiDAR quality with a handheld device able to perform quick captures, with harmless sensing technology, and an affordable price.
Also, we clarified in the prior sentence that the hardware version we are using is the old one.
Reviewer's comment 3:
What does the BRemNet stand for?
Thanks so much for pointing this omission, we added the following clarification at the very beginning of section 2:
We present BRemNet (Background Removal Network) ...